# CONTINUOUS AUTOREGRESSIVE LANGUAGE MODELS

## ABSTRACT

The efficiency of large language models (LLMs) is fundamentally limited by their sequential, token-by-token generation process. We argue that overcoming this bottleneck requires a new design axis for LLM scaling: increasing the semantic bandwidth of each generative step. To this end, we introduce Continuous Autoregressive Language Models (CALM), a paradigm shift from discrete next-token prediction to continuous next-vector prediction. CALM uses a high-fidelity autoencoder to compress a chunk of K tokens into a single continuous vector, thereby reducing the number of generative steps K-fold. This paradigm shift necessitates a new modeling toolkit; therefore, we develop a comprehensive likelihood-free framework that enables robust training, evaluation, and controllable sampling in the continuous domain without access to explicit probabilities. Experiments show that CALM significantly improves the performance-compute trade-off, achieving the performance of strong discrete baselines at a significantly lower computational cost. More importantly, these findings establish next-vector prediction as a powerful and scalable pathway towards ultra-efficient language models.

## 1 INTRODUCTION

Large Language Models (LLMs) have revolutionized the field of artificial intelligence, demonstrating unprecedented capabilities in understanding, generating, and reasoning with human language (Achiam et al., 2023; Google, 2025; DeepSeek-AI, 2025). However, this remarkable success is shadowed by a critical challenge: their immense computational demands. The training and inference of state-of-the-art LLMs demand massive computational resources, leading to prohibitive expenses and significant environmental concerns (Strubell et al., 2019; Bender et al., 2021). At the heart of this inefficiency lies the foundational paradigm of these models: an autoregressive generation process that operates on a sequence of discrete tokens. Because the computational cost scales with the length of the sequence, generating long-form text or processing extensive contexts remains a fundamental bottleneck, limiting the scalability and accessibility of these powerful models.

The now-ubiquitous use of discrete tokens in LLMs is the result of a pivotal evolution from earlier modeling paradigms. Initially, models that operated at the character level struggled with the computational burden of extremely long sequences (Sutskever et al., 2011; Kim et al., 2016). The subsequent shift to modern subword tokenization (Sennrich et al., 2016) was driven by a crucial insight: increasing the information density of each text unit reduces sequence length and dramatically boosts model efficiency. This historical success suggests a clear path for unlocking the next order of magnitude in efficiency: continue to increase the semantic bandwidth of each predictive unit.

We argue, however, that this path has reached a fundamental limit, constrained by the very nature of discrete representation. With typical vocabularies in modern LLMs ranging from approximately 32,000 to 256,000 entries, each token carries a surprisingly small amount of information—merely 15 to 18 bits (e.g., $\log_2(32768) = 15$). To increase this capacity—for instance, to represent a whole phrase—the vocabulary size would need to grow exponentially, making the final softmax computation over this vocabulary an untenable bottleneck. This reveals a critical limitation: the information density of discrete tokens is not scalable. Consequently, a profound mismatch has emerged: while model capacity has scaled to unprecedented levels, the task itself—predicting low-information discrete units one at a time—has not evolved. We are now deploying models of immense representational power on a task that fundamentally limits their throughput, forcing them to laboriously predict simple, low-information tokens one by one.

**Autoencoder (K=3 tokens to 1 vector)**

| The | cat | sat | on | the | mat |

| Vector 1 | Vector 2 |

Sequence Length = T                    Sequence Length = T/K

**Conventional LM: Next-Token Prediction**       **CALM: Next-Vector Prediction**

Figure 1: Comparison between conventional token-by-token generation and our proposed vector-by-vector framework (CALM). By compressing K tokens into a single vector, we reduce the sequence length K-fold, fundamentally improving computational efficiency.

In this work, we confront this limitation directly by introducing a paradigm shift from discrete tokens to a continuous-domain representation. Central to our approach is an autoencoder trained to compress a chunk of K tokens into a single, dense continuous vector and, crucially, reconstruct the original tokens from this vector with high fidelity. Unlike the discrete paradigm, where increasing information density requires an exponential growth in vocabulary size, our continuous representation offers a scalable path forward: the vector's information capacity can be gracefully expanded by simply increasing its dimensionality to accommodate a larger K. This design directly reduces the number of autoregressive steps by a factor of K. Ultimately, it allows us to reframe language modeling from a task of next-token prediction on discrete token sequences to next-vector prediction on continuous vector sequences, as conceptually illustrated in Figure 1.

However, shifting to the continuous domain introduces a significant challenge: without a finite vocabulary, a model cannot compute an explicit probability distribution over all possible outcomes using a standard softmax layer. To address this, we develop a comprehensive, likelihood-free framework for our Continuous Autoregressive Language Models (CALM). Our primary contributions, which structure the remainder of this paper, are as follows:

- **A Powerful and Lightweight Autoencoder (Section 2):** We first introduce an efficient autoencoder architecture designed to produce robust vector representations. We demonstrate that this model can be both compact and powerful, ensuring high-fidelity reconstruction of the original tokens, which is a prerequisite for the downstream language modeling task.

- **Likelihood-Free Language Modeling (Section 3):** To perform generative modeling in the continuous vector space, we employ a lightweight generative head that conditions on the last hidden state to generate the output vector. While the generative head can be any continuous generative model, options like Diffusion (Ho et al., 2020; Li et al., 2024) or Flow Matching (Lipman et al., 2023) rely on an iterative sampling process, re-introducing a significant inference bottleneck. Our framework therefore specifically adopts the Energy Transformer (Shao et al., 2025b), a recent architecture designed for efficient, single-step generation of continuous vectors, while empirically demonstrating superior generation quality.

- **Likelihood-Free LM Evaluation (Section 4):** The absence of explicit likelihoods makes traditional metrics like Perplexity inapplicable. We address this by proposing BrierLM, a novel metric for language modeling based on the Brier score (Brier, 1950). We show that BrierLM is strictly proper, theoretically ensuring a fair comparison of model capabilities. Crucially, BrierLM can be estimated unbiasedly by only drawing samples from the model, making it perfectly suited for CALM where likelihoods are intractable.

- **Likelihood-Free Temperature Sampling (Section 5):** Controlled generation via temperature sampling is an indispensable feature of modern LLMs, yet it relies on the explicit manipulation of a probability distribution. We introduce a principled, likelihood-free sampling algorithm that can, in theory, draw samples from the exact temperature distribution, and we accompany it with a highly efficient batch approximation.

We empirically validate our CALM framework on standard language modeling benchmarks, which demonstrates a superior performance-compute trade-off. For instance, a CALM grouping K=4 tokens delivers performance comparable to strong discrete baselines, but at a significantly lower computational cost. This findings highlight a new design axis for language models: rather than solely scaling parameters and data for performance, one can now scale the information capacity of each step as a powerful new lever for computational efficiency.

## 2 AUTOENCODER

### 2.1 HIGH-FIDELITY RECONSTRUCTION

The foundational component of our CALM framework is an autoencoder tasked with learning a bijective mapping between a chunk of $K$ discrete tokens and a continuous vector. Formally, we seek an encoder $f_{enc} : \mathcal{V}^K \to \mathbb{R}^l$ and a decoder $g_{dec} : \mathbb{R}^l \to \mathcal{V}^K$, where $\mathcal{V}$ is the vocabulary, such that for a given token sequence $\mathbf{x}_{1:K} = (x_1, \ldots, x_K)$, the reconstruction $g_{dec}(f_{enc}(\mathbf{x}_{1:K}))$ closely approximates $\mathbf{x}_{1:K}$. For simplicity and computational efficiency, we design our autoencoder to be *context-free*, meaning it processes each token chunk independently of its surrounding sequence. A context-aware autoencoder that also conditions on previous vector representations is a natural and promising next step, which we leave for future exploration.

The encoder begins by mapping the input sequence $\mathbf{x}_{1:K}$ to $K$ embeddings. Each embedding is independently processed by a position-wise feed-forward network (FFN). The resulting $K$ hidden states are then flattened and compressed by a linear layer: $\mathbb{R}^{Kd} \to \mathbb{R}^d$. This unified representation is passed through a second FFN and a linear projection to produce the $l$-dimensional latent vector $\mathbf{z}$.

The decoder architecture mirrors the encoder. It first transforms $\mathbf{z}$ using a linear layer and an FFN to obtain a $d$-dimensional hidden state, which is then expanded by another linear layer to dimension $Kd$ and reshaped into a sequence of $K$ hidden states. Each of these states is passed through a second FFN, followed by a projection to vocabulary logits using the tied input embedding matrix. Finally, the tokens are reconstructed by applying an argmax operation over these logits.

The autoencoder is trained to minimize the reconstruction error by optimizing the standard cross-entropy loss across all $K$ token positions:

$$\mathcal{L}_{ae}(\mathbf{x}_{1:K}) = -\sum_{i=1}^{K} \log p_{dec}(x_i | \mathbf{z} = f_{enc}(\mathbf{x}_{1:K})). \tag{1}$$

We empirically validate this architecture and find it to be both highly effective and efficient. For instance, when grouping $K = 4$ tokens, a latent vector of just $l = 10$ dimensions is sufficient to achieve high-fidelity reconstruction, with a token-level accuracy of over 99.9%. Moreover, the autoencoder is exceptionally lightweight; with a shallow architecture and a modest hidden dimension of $d = 512$, its computational overhead is nearly negligible compared to that of language model.

### 2.2 ROBUST VECTOR REPRESENTATION

While the autoencoder described above achieves near-perfect reconstruction, we found that it is practically impossible to effectively train a continuous language model based on the vector space it produces. The root cause of this challenge is that an autoencoder optimized solely for reconstruction learns an exceptionally brittle representation. Lacking any incentive to form a smooth latent manifold, the encoder learns to pack information with maximum efficiency, creating a highly irregular mapping. In such a space, a minor perturbation to a latent vector $\mathbf{z}$—such as the small, inevitable errors made by a generative can cause the decoder to reconstruct a completely unrelated token sequence. Therefore, for our CALM framework to be viable, the autoencoder must satisfy another critical objective: its vector representation should be robust.

**Variational Regularization.** To build a robust latent space, our primary strategy is to smooth the latent manifold by moving from a deterministic autoencoder to a variational one (Kingma & Welling, 2014), aligning our approach with prominent generative models (Rombach et al., 2022; Liu et al., 2023) that operate within a smooth and structured latent space. Instead of mapping an input chunk

directly to a vector $\mathbf{z}$, the encoder now outputs the parameters of a diagonal Gaussian distribution, $\boldsymbol{\mu}$ and $\boldsymbol{\sigma}$, from which the latent vector is sampled: $\mathbf{z} \sim \mathcal{N}(\boldsymbol{\mu}, \boldsymbol{\sigma}^2 \mathbf{I})$. This change is accompanied by a new objective term, a KL divergence loss that penalizes the deviation of the encoded distribution from a standard normal prior, $\mathcal{N}(0, \mathbf{I})$. The total loss function is thus a weighted sum of the reconstruction and regularization terms:

$$\mathcal{L}_{\text{total}} = \mathcal{L}_{\text{ae}} + \beta \cdot \mathcal{L}_{\text{KL}}, \tag{2}$$

where $\beta$ is a hyperparameter balancing the two objectives (we set $\beta = 0.001$), and $\mathcal{L}_{\text{KL}}$ is the KL divergence, defined as:

$$\mathcal{L}_{\text{KL}}(p_E(\mathbf{z}|\mathbf{x}_{1:K}) \| \mathcal{N}(0, \mathbf{I})) = -\frac{1}{2} \sum_{i=1}^{l} (1 + \log \sigma_i^2 - \sigma_i^2 - \mu_i^2). \tag{3}$$

This variational objective discourages the encoder from relying on arbitrarily precise or large-magnitude values in $\mathbf{z}$, thereby promoting a smoother and more regularized latent manifold that is more amenable to generative modeling.

**Preventing Posterior Collapse.** A significant challenge in training VAEs is posterior collapse. This issue manifested in our model as a tendency for some latent dimensions to fully collapse to the standard normal prior. While collapsing a dimension drives its KL divergence to zero, it renders that dimension uninformative for reconstruction. More critically, these pure noise dimensions introduce a chaotic signal that interferes with the training of the downstream language model, destabilizing the learning process. To mitigate this, we adopt the KL clipping strategy from Kingma et al. (2016), which modifies the objective by clipping each dimension's KL loss at a constant floor:

$$\mathcal{L}_{\text{KL}}^{clip} = \sum_{i=1}^{l} \max(\lambda_{KL}, \mathcal{L}_{\text{KL},i}), \tag{4}$$

where $\mathcal{L}_{\text{KL},i}$ is the KL divergence for the $i$-th dimension and $\lambda_{KL}$ is the threshold (we use $\lambda_{KL} = 0.5$). This technique ensures that every dimension is encouraged to actively participate in reconstruction, thus preventing collapse and fostering a dense, structured representation.

**Dropout for Enhanced Robustness.** Beyond structuring the latent space with variational methods, we further enhance its robustness by injecting noise during training using two complementary forms of dropout. First, we apply dropout with a rate of $p = 0.15$ to the latent vector $\mathbf{z}$ before it is passed to the decoder. This forces the autoencoder to learn a redundant representation, making it robust to minor prediction errors from the downstream generative model. Second, we apply dropout to input tokens by randomly masking a fraction ($p = 0.15$) of tokens. Analogous to the Continuous Bag-of-Words (CBOW) method (Mikolov et al., 2013), this compels the autoencoder to infer masked tokens from their context, thereby enriching the latent vector with the chunk's semantic context rather than just performing a simple token-index compression. Crucially, these dropout techniques are employed exclusively during the autoencoder's training phase to build a robust latent representation; they are disabled during the subsequent training and inference of the continuous language model.

The synthesis of these techniques produces a powerful and robust autoencoder. For a chunk of $K = 4$ tokens, we now employ a latent vector of $l = 128$ dimensions, providing the necessary capacity to encode information redundantly. The encoder learns a posterior distribution where the standard deviations, $\sigma_i$, converge to approximately 0.3. This means that sampling the latent vector $\mathbf{z}$ effectively perturbs the predicted mean $\boldsymbol{\mu}$ with a substantial Gaussian noise $\boldsymbol{\sigma} \approx 0.3\mathbf{I}$. Despite this significant latent perturbation, the decoder still maintains a token-level accuracy exceeding 99.9%. This vector representation, which combines high fidelity with high robustness, lays a solid foundation for the subsequent learning of Continuous Autoregressive Language Models (CALM).

## 3 Likelihood-Free Language Modeling

### 3.1 Next-Vector Prediction

The autoencoder developed in Section 2 establishes a robust and high-fidelity mapping between a chunk of $K$ discrete tokens and a single continuous vector, which allow us to reframe language

modeling from a task of next-token prediction on discrete token sequences to next-vector prediction on continuous vector sequences. Specifically, a sequence of $T$ tokens, $\mathbf{X} = (x_1, \ldots, x_T)$, is first grouped into $L = T/K$ non-overlapping chunks. The encoder, $f_{\text{enc}}$, then transforms the original sequence into a new, more compact sequence of continuous vectors:

$$\mathbf{Z} = (\mathbf{z}_1, \mathbf{z}_2, \ldots, \mathbf{z}_L), \quad \text{where} \quad \mathbf{z}_i = f_{\text{enc}}(x_{(i-1)K+1}, \ldots, x_{iK}). \tag{5}$$

Consequently, the autoregressive objective evolves to predicting the next vector in the sequence:

$$p(\mathbf{Z}) = \prod_{i=1}^{L} p(\mathbf{z}_i | \mathbf{z}_{<i}). \tag{6}$$

While this autoregressive structure is preserved, the underlying mechanism for predicting the next element must be redesigned. Unlike standard language models, which rely on a softmax layer to compute a probability distribution over a finite vocabulary, our model must predict a vector within the infinite space $\mathbb{R}^l$. The softmax function is not applicable over this uncountable set, rendering the explicit probability density $p(\mathbf{z}_i | \mathbf{z}_{<i})$ intractable. This introduces two critical challenges:

- **Training:** The likelihood $p(\mathbf{z}_i | \mathbf{z}_{<i})$ becomes intractable, precluding the use of maximum likelihood estimation (i.e., minimizing cross-entropy loss) for training.
- **Evaluation:** Standard evaluation metrics like Perplexity, which are derived directly from the model's likelihood, can no longer be computed to measure model performance.

We address both of these challenges in turn. For the training problem, we introduce our approach to likelihood-free language modeling in the remainder of this section. For the evaluation problem, we propose a likelihood-free evaluation methodology in Section 4.

## 3.2 GENERATIVE HEAD

Generative modeling of continuous data (Kingma & Welling, 2014; Goodfellow et al., 2014; Ho et al., 2020) is a well-established field, foundational to domains such as image and audio synthesis where data is inherently continuous. A promising recent paradigm (Tschannen et al., 2023; Li et al., 2024; Shao et al., 2025b) combines these approaches with autoregressive models: a Transformer backbone predicts a conditioning hidden state, which is used by a subsequent generative model to produce the continuous output for each step. Our Continuous Autoregressive Language Models (CALM) adapts this paradigm, but with a critical focus on computational efficiency that constrains the design of this generative component. We therefore conceptualize this component as a lightweight *generative head*. Formally, the generative head is a stochastic function that takes the Transformer's hidden state, $\mathbf{h}_{i-1} \in \mathbb{R}^d$, and draws a sample $\mathbf{z}_i \in \mathbb{R}^l$ from the conditional distribution:

$$\mathbf{h}_{i-1} = \text{Transformer}(\mathbf{z}_{1:i-1}), \quad \mathbf{z}_i \sim p(\cdot | \mathbf{h}_{i-1}). \tag{7}$$

While the generative head can be any continuous generative model, prominent options like Diffusion (Ho et al., 2020; Li et al., 2024; Fan et al., 2025) or Flow Matching (Lipman et al., 2023; Ren et al., 2025a;b) are misaligned with our goal of efficiency. These models rely on an iterative sampling process—requiring dozens or even hundreds of network evaluations to produce a single vector—which directly counteracts the speedup gained from reducing the number of autoregressive steps. The CALM architecture therefore demands a generative head capable of high-quality, single-step generation, a challenge we address next with an energy-based objective.

## 3.3 ENERGY TRANSFORMER

### 3.3.1 STRICTLY PROPER SCORING RULES

To meet the demand for a generative head capable of high-quality, single-step generation, we draw inspiration from Shao et al. (2024; 2025b), which frames the generative task as the optimization of strictly proper scoring rules (Gneiting & Raftery, 2007). Formally, a scoring rule $S(P, y)$ assigns a numerical score to a predictive distribution $P$ upon observing an outcome $y$, where higher scores are better. The quality of a forecast $P$ against the true data-generating distribution $Q$ is measured by

its expected score, defined as $S(P, Q) = \mathbb{E}_{y \sim Q}[S(P, y)]$. A scoring rule is considered *proper* if the expected score is maximized when the predictive distribution $P$ matches the data distribution $Q$:

$$S(P, Q) \leq S(Q, Q) \quad \text{for all distributions } P. \tag{8}$$

This property ensures that the scoring rule does not incentivize the model to predict a biased or distorted distribution. Furthermore, a scoring rule is *strictly proper* if equality holds only when $P = Q$, meaning that the optimal score can only be achieved by reporting the true distribution.

The use of a strictly proper scoring rule as a training objective is therefore a powerful and principled approach for training our generative head, as maximizing the expected score is equivalent to driving the model's predictive distribution to match the true distribution. This principle offers a direct generalization of maximum likelihood estimation, where the negative log-likelihood is a special case corresponding to the logarithmic score (Good, 1952). While the likelihood is intractable in the continuous domain, the theory of scoring rules provides a rich family of alternatives.

### 3.3.2 ENERGY LOSS

We build our training objective using the Energy Score (Székely, 2003), a strictly proper scoring rule that has proven effective across a range of generative applications (Gritsenko et al., 2020; Vahidi et al., 2024; Pacchiardi et al., 2024; Shao et al., 2025b; Ma et al., 2025). The energy score is entirely likelihood-free; rather than evaluating probability densities, it measures the alignment between the prediction and the observation via sample distances. For a predictive distribution $P$ and a ground truth observation $\mathbf{y}$, the energy score is defined as:

$$S(P, \mathbf{y}) = \mathbb{E}_{\mathbf{x}', \mathbf{x}'' \sim P}[\|\mathbf{x}' - \mathbf{x}''\|^\alpha] - 2\,\mathbb{E}_{\mathbf{x} \sim P}[\|\mathbf{x} - \mathbf{y}\|^\alpha], \tag{9}$$

where $\mathbf{x}$, $\mathbf{x}'$ and $\mathbf{x}''$ are independent samples drawn from $P$. The score is strictly proper for any $\alpha \in (0, 2)$. Typically, $\alpha$ is set to 1. The first term encourages diversity, penalizing the model for producing collapsed or overly confident predictions where all samples are identical. The second term encourages fidelity, driving the model's predictions to be close to the ground truth observation.

While the expectations in Equation 9 make the energy score intractable to compute exactly, we can construct an unbiased Monte Carlo estimator to serve as a practical loss function, which we term the *energy loss*. To do this, we draw $N$ candidate samples, $\{\tilde{\mathbf{z}}_{i,1}, \ldots, \tilde{\mathbf{z}}_{i,N}\}$, from the generative head at each step $i$. Furthermore, we leverage a unique property of our setup: our autoencoder does not map a token chunk to a fixed point, but rather to a conditional Gaussian posterior $\mathbf{z}_i \sim q(\cdot | \mathbf{x}_{(i-1)K+1:iK})$. Relying on a single sample $\mathbf{z}_i$ as ground-truth can introduce high variance into the energy loss. To mitigate this and stabilize training, we draw $M$ target samples, $\{\mathbf{z}_{i,1}, \ldots, \mathbf{z}_{i,M}\}$, from this posterior. Combining these sample sets, the final energy loss is formulated as:

$$\mathcal{L}_{\text{energy}} = \sum_{i=1}^{L} \left( \frac{2}{NM} \sum_{n=1}^{N} \sum_{m=1}^{M} \|\mathbf{z}_{i,m} - \tilde{\mathbf{z}}_{i,n}\| - \frac{1}{N(N-1)} \sum_{n \neq k} \|\tilde{\mathbf{z}}_{i,n} - \tilde{\mathbf{z}}_{i,k}\| \right). \tag{10}$$

In practice, we set $N = 8$ and $M = 100$. The number of model samples $N$ directly scales the training cost, as each sample requires an evaluation of the generative head; we therefore use a small $N$ to maintain high training efficiency. The overhead of drawing target vectors from a known Gaussian posterior is almost negligible, which allows us to use a large $M$ to reduce the variance of loss.

A key advantage of this likelihood-free training objective is its flexibility: it only requires the ability to draw samples from the generative head, placing minimal constraints on its internal architecture and allowing for the simple and efficient designs we explore next.

### 3.3.3 MODEL ARCHITECTURE

We now detail our model architecture. We use a standard Transformer backbone, with modifications focused on the output-side generative head and the input-side adaptation.

**Energy-Based Generative Head**. The inputs to the generative head are twofold: the hidden state $\mathbf{h}_{i-1}$ from the Transformer backbone, which provides the conditional context, and a random noise vector $\boldsymbol{\varepsilon} \in \mathbb{R}^{d_{\text{noise}}}$, which provides the necessary stochasticity for sampling. Each dimension of $\boldsymbol{\varepsilon}$ is drawn independently from a uniform distribution $\mathcal{U}[-0.5, 0.5]$. Both the hidden state $\mathbf{h}_{i-1}$ and the

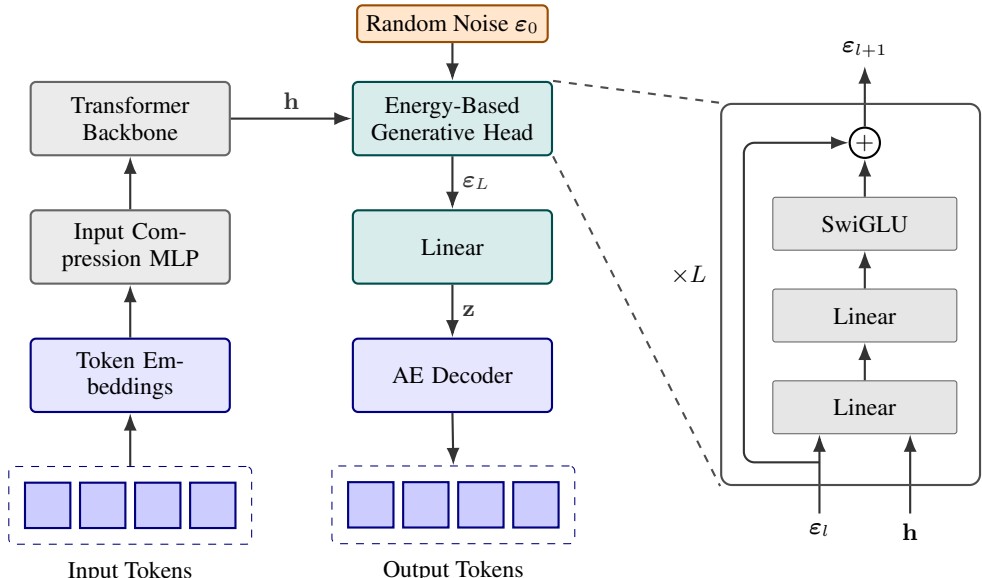

Figure 2: The Architecture of the Continuous Autoregressive Language Model (CALM). Left: The main autoregressive loop where discrete tokens are compressed to condition a Transformer, whose output hidden state $\mathbf{h}$ guides an energy-based head to predict a continuous vector $\mathbf{z}$. The AE decoder then maps $\mathbf{z}$ back to discrete tokens for the next step. Right: A detailed view of the generative head, showing how it refines a noise vector $\varepsilon_0$ through a series of residual MLP blocks.

noise vector $\varepsilon$ are projected by independent linear layers to match the head's internal dimension, which we set to match the Transformer's hidden dimension $d$.

The core of the generative head is a stack of $L$ residual MLP blocks that progressively refine the initial noise representation $\varepsilon_0 = \varepsilon$ into the final output vector. As illustrated in Figure 2, each MLP block first fuses the current representation $\varepsilon_l$ with the hidden state via two linear layers. This is followed by a SwiGLU layer (Shazeer, 2020b) with an intermediate dimension of $d$. A residual connection then adds the block's input to its output. This process concludes with a final linear layer that projects the representation to the target dimension $l$, producing the output vector $\mathbf{z}_i$.

A single MLP block contains approximately $6d^2$ parameters. We set the number of blocks to a quarter of the number of Transformer layers; the entire generative head therefore accounts for only about $10\%$ of the total model parameters, making its computational overhead minimal.

**Discrete Token Input**. An intuitive approach for the model's input would be to embed the predicted latent vectors $\mathbf{z}_{i-1}$ from the previous step into the Transformer's hidden dimension $d$ using a linear projection. However, we empirically found that using these latent vectors as input for the Transformer leads to a noticeable degradation in performance, as the model struggles to unpack the semantic information from such a compact and brittle input representation.

To circumvent this, we ground the model's autoregressive process in the discrete token space. During training, the input for each step is formed by the $K$ tokens from the previous step. To maintain efficiency, we use a lightweight input compression module—a two-layer MLP—to map the K embeddings into a single input representation. The inference process unfolds as follows:

1. **Input Processing:** At step $i$, the previously generated chunk of $K$ tokens are embedded and compressed into a single input representation and fed into the Transformer.

2. **Continuous Prediction:** The Transformer outputs the hidden state $\mathbf{h}_{i-1}$, which our energy-based generative head then uses to predict the next continuous vector, $\mathbf{z}_i$.

3. **Discrete Feedback Loop:** The predicted vector $\mathbf{z}_i$ is immediately passed through the frozen decoder of our pre-trained autoencoder, $g_{\text{dec}}$, to reconstruct the next $K$ discrete tokens.

The complete architecture of CALM is illustrated in Figure 2.

## 4 Likelihood-Free LM Evaluation

Standard metrics like Perplexity are inapplicable to our likelihood-free framework. To address this, we introduce BrierLM, a novel evaluation metric based on the Brier score—a strictly proper scoring rule that guarantees a fair assessment of a model's predictive distribution (Brier, 1950). Our key contribution is an unbiased Monte Carlo estimator that computes the Brier score using only model samples, making it universally applicable to CALM, traditional LLMs, and other models. Empirical results demonstrate a strong agreement between BrierLM and Perplexity, confirming its reliability as an evaluation metric. Due to space constraints, we leave the detailed discussion in Appendix A.

## 5 Likelihood-Free Temperature Sampling

Temperature sampling is a critical feature for modern LLMs, but conventional methods rely on rescaling output logits, which is incompatible with our framework. We overcome this by developing a provably exact rejection sampling algorithm that performs temperature sampling using only a black-box sampler. We also accompany it with an efficient, asymptotically unbiased batch approximation. The formal algorithms and their theoretical analyses are detailed in Appendix B.

## 6 Experiments

### 6.1 Settings

**Datasets.** We train our models on the Pile uncopyrighted dataset (Gao et al., 2020). The raw text is processed with the Llama 3 tokenizer (Grattafiori et al., 2024), resulting in a training set of $\sim$230B tokens. We evaluate model performance on the WikiText-103 benchmark (Merity et al., 2017).

**Model.** Our models are built upon a standard Transformer backbone. We adopt most of the architecture designs from the LLaMA family (Touvron et al., 2023), including RMSNorm (Zhang & Sennrich, 2019), SwiGLU activation (Shazeer, 2020a), and rotary positional embeddings (Su et al., 2021). We experiment with four scales: S (12 layers, hidden_size=768, intermediate_size=2048), M (16 layers, hidden_size=1024, intermediate_size=2752), L (16 layers, hidden_size=1536, intermediate_size=4096), and XL (16 layers, hidden_size=2560, intermediate_size=6880).

**Training Details.** The training process for our CALM framework is two-staged. We first train a suite of autoencoders on a 15B token subset of the Pile to map token chunks of size $K \in \{1, 2, 4, 8\}$ into continuous vectors. These autoencoders use a hidden size of 512, a latent dimension of $32K$, have approximately 75M parameters, and are trained for 30k steps with a batch size of 512k tokens. Following this, the CALM models are trained on the remaining data for 250k steps with a batch size of 2 million tokens. The context length is set to 2048 steps; for CALM, this corresponds to $2048K$ tokens. All models are optimized using the AdamW optimizer (Loshchilov & Hutter, 2019) with $\beta_1 = 0.9, \beta_2 = 0.95, \epsilon = 1e - 8$. We use a learning rate of $3 \times 10^{-4}$ with a constant schedule and a warmup of 2000 steps, a weight decay of 0.1, and gradient clipping of 1.0.

### 6.2 Main Results

We present the primary results of our comparison between the standard Transformer baselines and our CALM framework (with a fixed chunk size of K=4) in Table 1. The results demonstrate that CALM establishes a new, more efficient performance-compute frontier for language modeling. By increasing the semantic bandwidth of each autoregressive step, CALM is allowed to be substantially larger in parameter count while demanding fewer FLOPs for both training and inference. For instance, our 371M parameter CALM-M model achieves a BrierLM score comparable to the 281M Transformer-S baseline, yet requires 44% fewer training FLOPs and 34% fewer inference FLOPs. Furthermore, the results confirm that CALM benefits from scaling just as effectively as traditional Transformers, allowing performance to be consistently improved by increasing model size.

In addition to scaling model size, our framework introduces the chunk size $K$ as a new lever for navigating the performance-compute landscape. Figure 3 illustrates this by plotting the performance of CALM-L with varying $K$ against the standard Transformer scaling curve. Notably, CALM-L with $K = 1$ operates at a significant disadvantage, demanding more FLOPs for lower performance

Table 1: Performance and computational cost comparison between Transformer baselines and CALM (K=4). CALM's reported parameter counts and FLOPs include all overhead from the autoencoder (75M parameters, training cost, and encoding/decoding FLOPs). Attention FLOPs are calculated assuming a context length of 2048.

| Model | #Params | Train FLOPs (total, 1e20) | Infer FLOPs (per token, 1e8) | BrierLM |
|---|---|---|---|---|
| Transformer-S | 281M | 6.6 | 4.4 | 6.05 |
| Transformer-M | 465M | 11.9 | 7.9 | 7.07 |
| Transformer-L | 849M | 22.5 | 15.0 | 8.98 |
| CALM-M (K=4) | 371M | 3.7 | 2.9 | 5.72 |
| CALM-L (K=4) | 735M | 7.7 | 4.6 | 6.58 |
| CALM-XL (K=4) | 1.82B | 19.5 | 9.4 | 8.53 |

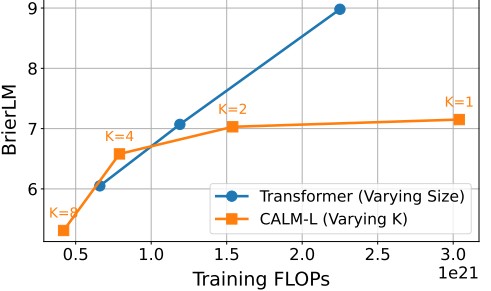

Figure 3: The effect of chunk size K on the performance-compute trade-off.

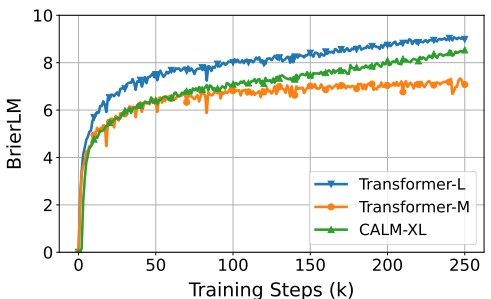

Figure 4: Training progress of CALM and traditional Transformer models.

compared to its discrete counterpart. The advantages of CALM become apparent as we increase $K$. Moving from $K = 1$ to $K = 2$ nearly halves the cost with only a marginal drop in performance, and at $K = 4$, the CALM model surpasses the baseline performance-compute frontier. This finding validates our central hypothesis: scaling the semantic bandwidth of each step provides a highly effective axis for optimizing the performance-compute trade-off. Further increasing the chunk size to 8 leads to a larger performance drop, which is likely a model capacity limitation. We hypothesize that larger models may be required to leverage the benefits of higher semantic bandwidths.

To further investigate the learning dynamics of our framework, we plot the training curves of CALM-XL against the Transformer baselines in Figure 4. The baseline Transformer models exhibit rapid initial gains before their performance gradually begins to saturate. In contrast, CALM-XL displays a more patient but ultimately steeper learning curve. We attribute this phenomenon to the different nature of the predictive task. While the baseline models learn the relatively simple task of predicting a single, low-information discrete token, our CALM model must learn to model the complex, high-dimensional distribution of continuous vectors, which explains the slower initial progress. However, once this ability is established, the model can unlock the potential of its large parameter count, entering a phase of more significant and sustained improvements.

Due to space constraints, we present some supplementary experimental results in Appendix D.

## 7 CONCLUSION

In this work, we challenge the inefficient, token-by-token paradigm of LLMs by introducing Continuous Autoregressive Language Models (CALM), a framework that shifts generation from discrete tokens to a continuous vector space where a single vector represents K tokens. To support this approach, we develop a comprehensive likelihood-free toolkit: a robust and high-fidelity autoencoder, the energy loss for generative modeling, the BrierLM metric for LM evaluation, and a new suite of algorithms for temperature sampling. Empirical results show that CALM achieves a superior performance-compute trade-off, highlighting a new scaling axis for language modeling: scaling the semantic bandwidth of each generative step to push the performance-compute frontier of LLMs.

## 8 REPRODUCIBILITY STATEMENT

To ensure the reproducibility of our research, we have submitted the source code for our CALM framework as supplementary materials. Upon publication, we intend to publicly release our model checkpoints, along with a detailed README.md file providing step-by-step guidance for the training and evaluation.

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

# A LIKELIHOOD-FREE LM EVALUATION

## A.1 PRINCIPLES OF LM EVALUATION

The CALM framework operates as an implicit generative model, whose predictive probability distribution is defined by its sampling process. Consequently, standard LM evaluation metrics like Perplexity, which are defined in terms of explicit likelihoods, can no longer be employed to measure model performance. Furthermore, the energy loss used for training is itself unsuitable for evaluation, as its magnitude is subjective to the specific latent space shaped by the autoencoder. This necessitates the development of a model-agnostic evaluation metric, one that can faithfully assess language modeling capabilities in a principled, yet entirely likelihood-free, manner.

The goal of a evaluation metric is to quantify the divergence between the model's predictive distribution, $P$, and the true data distribution, $Q$. This principle is formalized by the property that the metric is uniquely optimized when the model accurately recovers the data distribution ($P = Q$). This ensures the evaluation is fair and cannot be hacked by a model that systematically distorts its predictions. For instance, the conventional metric of Perplexity serves as a prime example of this principle. It is grounded in the expected negative log-likelihood, which can be decomposed into the sum of the KL divergence and data entropy:

$$\mathbb{E}_{y\sim Q}[-\log P(y)] = \mathbb{E}_{y\sim Q}\left[\log \frac{Q(y)}{P(y)}\right] + \mathbb{E}_{y\sim Q}[-\log Q(y)] = \underbrace{D_{KL}(Q\|P)}_{\text{Minimized at } P=Q} + \underbrace{H(Q)}_{\text{Constant}}. \quad (11)$$

This property establishes Perplexity as a theoretically sound measure of a model's capability to capture the true distribution, which is uniquely minimized when $P = Q$.

In contrast, a naive metric like the raw likelihood of the observed outcome, $P(y)$, fails this principle. The expected score under this metric, $\mathbb{E}_{y\sim Q}[P(y)]$, is maximized by a deterministic prediction that assigns a probability of 1 to the single most frequent outcome, i.e., $P(\arg\max_y Q(y)) = 1$. Such a metric would therefore incorrectly favor an overconfident model that fails to capture the underlying data uncertainty. This highlights a critical distinction: a principled metric must balance rewarding accuracy with correctly representing the predictive uncertainty. The naive likelihood $P(y)$ only addresses the former, making it an inadequate measure of a model's predictive quality.

## A.2 BRIERLM: BRIER FOR LANGUAGE MODELING

For a principled and likelihood-free evaluation, we turn to the Brier score (Brier, 1950), a classic strictly proper scoring rule now widely used to assess the calibration of modern neural networks (Lakshminarayanan et al., 2017; Ovadia et al., 2019; Gruber & Buettner, 2022). For a predictive distribution $P$ and a ground-truth outcome $y$, the Brier score is defined as:

$$\text{Brier}(P, y) = 2P(y) - \sum_x P(x)^2. \quad (12)$$

Unlike the raw likelihood $P(y)$, which solely measures accuracy, the Brier score incorporates an additional term, $\sum_x P(x)^2$, to quantify predictive uncertainty. This structure balances two competing objectives, which ultimately rewards a well-calibrated prediction. This property is revealed by the following decomposition of the expected Brier score:

$$\mathbb{E}_{y\sim Q}[\text{Brier}(P, y)] = -\underbrace{\sum_x (P(x) - Q(x))^2}_{\text{Squared Error (minimized at } P=Q)} + \underbrace{\sum_x Q(x)^2}_{\text{Data Variance (constant)}}. \quad (13)$$

While the Brier score is theoretically sound, its direct computation remains intractable for CALM, as it requires knowledge of the full predictive distribution $P$. We find, however, that an unbiased Monte Carlo estimator for the Brier score can be constructed in an entirely likelihood-free manner, using only samples drawn from the model. Specifically, the uncertainty term, $\sum_x P(x)^2$, can be interpreted as the collision probability of two independent samples. Therefore, its unbiased estimator is simply the indicator function $\mathbb{I}\{x_1 = x_2\}$, where $x_1, x_2 \sim P$. Similarly, the accuracy term $P(y)$

can be estimated by $\mathbb{I}\{x = y\}$ using a single sample $x \sim P$. Combining these, we construct a practical, unbiased estimator for the Brier score using two samples drawn from the model:

$$\text{Brier}(P, y) \approx \mathbb{I}\{x_1 = y\} + \mathbb{I}\{x_2 = y\} - \mathbb{I}\{x_1 = x_2\}, \quad x_1, x_2 \sim P. \tag{14}$$

This estimator enables a likelihood-free evaluation of CALM's predictive capabilities. A straightforward approach is to assess next-token prediction performance in a teacher-forcing setting. This would involve generating two latent vectors at each step, decoding them using the frozen autoencoder's decoder, and computing the Brier score using only the first token of each resulting chunk. However, such an evaluation is insufficient as it ignores the generation quality of the remaining $K - 1$ tokens. To address this limitation, we further introduce *Brier-n*, a metric that computes the Brier score over entire n-grams. In this formulation, the indicator functions of the estimator treat the n-gram as a single, atomic outcome. Finally, following the convention of established n-gram-based metrics like BLEU (Papineni et al., 2002), we define our composite metric, *BrierLM* (Brier for Language Modeling), as the geometric mean of Brier-$n$ scores for $n = 1$ to $4$, which we then scale by 100 to place it on a more interpretable 0-100 range:

$$\text{BrierLM} = 100 \cdot \left( \prod_{n=1}^{4} \text{Brier-}n \right)^{0.25}. \tag{15}$$

The utility of BrierLM extends beyond CALM, serving as a universal evaluation protocol that is also applicable to conventional autoregressive models. For such models, the BrierLM estimator can be applied by simply drawing samples from the final softmax distribution, enabling direct and fair comparisons with our likelihood-free framework. To validate this, we evaluated both cross-entropy and BrierLM throughout the training of our baseline autoregressive models (detailed in Section 6.1). Figure 5 visualizes the joint distribution of the two metrics. Intriguingly, we find that BrierLM is highly consistent with cross-entropy loss, exhibiting a nearly linear relationship with a Pearson correlation coefficient of -0.966 and a Spearman's rank correlation of -0.991. This strong monotonic alignment confirms that BrierLM is a reliable measure of language modeling capability, establishing it as a trustworthy likelihood-free alternative to Perplexity.

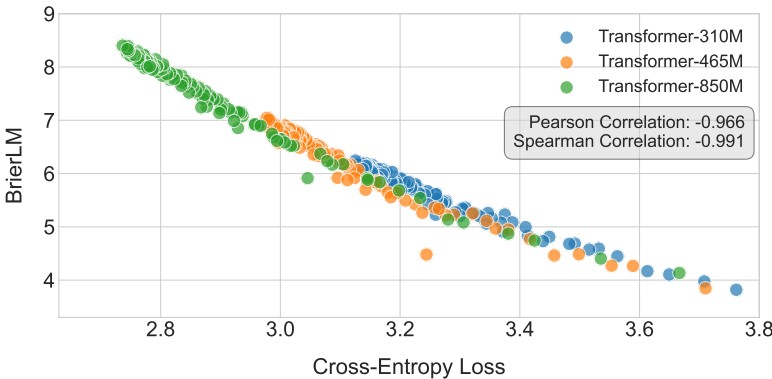

Figure 5: Joint distribution of the cross-entropy loss and the BrierLM score across different models and training checkpoints.

Furthermore, BrierLM offers a particularly significant advantage for the growing class of implicit generative models, such as diffusion-based language models (Austin et al., 2021; Han et al., 2023; Lou et al., 2024; Arriola et al., 2025). These models have historically been challenging to evaluate, often relying on the complex and sometimes loose estimation of variational lower bounds (ELBOs) to approximate Perplexity. BrierLM circumvents this entire challenge, offering a direct, unbiased method to faithfully assess their language modeling capabilities and enabling fair comparisons across different model families.

---

**Algorithm 1** Likelihood-free Temperature Sampling

---

**Input:** A base sampler $S$ for an implicit discrete distribution $P(x)$; A target temperature $T \in (0, 1)$
**Output:** Sample $x$ accepted with probability $P_T(x) \propto P(x)^{1/T}$

1: **procedure** SAMPLEATTEMPERATURE($S, T$)
2:     $n \leftarrow \lfloor 1/T \rfloor$.                                           ▷ Integer part of $1/T$
3:     $\alpha \leftarrow 1/T - n$.                                           ▷ Fractional part, $0 \leq \alpha < 1$
4:     **Stage 1: Integer Part** ($n$)
5:     Draw $n$ i.i.d. samples $x_1, \ldots, x_n \sim S$
6:     **if** $x_1 = \cdots = x_n$ **then**
7:         $x^* \leftarrow x_1$                                           ▷ Find candidate $x^*$
8:     **else**
9:         **restart from stage 1**                                           ▷ Rejection
10:     **if** $\alpha = 0$ **then**
11:         **return** $x^*$                                           ▷ Accept $x^*$ as Stage 2 is not needed
12:     **Stage 2: Fractional Part** ($\alpha$)
13:     $i \leftarrow 1$
14:     **loop**
15:         Draw $x \sim S$
16:         **if** $x = x^*$ **then**
17:             **return** $x^*$                                           ▷ Accept candidate
18:         **else**
19:             Draw $u \sim \mathcal{U}(0, 1)$                                           ▷ Uniform distribution
20:             **if** $u < \alpha/i$ **then**
21:                 **restart from stage 1**                                           ▷ Rejection
22:             **else**
23:                 $i \leftarrow i + 1$                                           ▷ Continue to next iteration

---

# B   LIKELIHOOD-FREE TEMPERATURE SAMPLING

## B.1   EXACT TEMPERATURE SAMPLING VIA REJECTION

Controlled generation via temperature sampling is an indispensable feature of modern LLMs. Conventionally, this technique is implemented by rescaling pre-softmax logits, a mechanism that requires explicit access to the model's probability distribution. However, this approach is incompatible with our CALM framework, whose generative head is likelihood-free and provides only a sampler. This presents a critical challenge: performing temperature sampling with only a black-box sampler. In this section, we address this challenge by developing an exact algorithm, grounded in the principles of rejection sampling, that provably achieves this goal.

The intuition for our algorithm stems from the relationship between repeated sampling and probability exponentiation. In the context of CALM, a sample $x$ corresponds to a complete chunk of $K$ tokens produced at each step. Consider the simple case where the temperature $T = 1/n$ for an integer $n$, which makes the target distribution $P_T(x) \propto P(x)^n$. The probability of drawing the exact same sample $x$ in $n$ independent trials from the sampler is also $P(x)^n$. This motivates an elegant rejection sampling scheme: we draw $n$ samples and accept them if and only if all $n$ samples are identical. Otherwise, we reject the entire set and restart the process. The distribution of accepted samples is thus provably proportional to $P(x)^n$, providing a foundation for our general algorithm.

To generalize this approach for any arbitrary temperature $T \in (0, 1)$, we decompose the exponent $1/T$ into its integer part, $n = \lfloor 1/T \rfloor$, and fractional part, $\alpha = 1/T - n$. This decomposition structures our algorithm as a two-stage rejection sampling process. The first stage handles the integer component $n$ using the repetition-based scheme described above, producing a candidate sample $x$ only if $n$ independent draws are identical. The second stage, which handles the fractional exponent $\alpha$, requires a more subtle approach. Here, we draw upon the theory of Bernoulli Factory (Keane & O'Brien, 1994; Mendo, 2019) to construct an iterative procedure that simulates a biased coin flip with a success probability of $P(x)^\alpha$. A sample is accepted only if it passes both stages; failure at any point triggers a restart of the entire process. The complete procedure is formally detailed in Algorithm 1. The following theorem guarantees its correctness.

**Theorem 1.** *For an implicit discrete distribution $P(x)$ with sampler $S$ and a temperature $T \in (0,1)$, Algorithm 1 generates samples distributed as:*

$$P_T(x) = \frac{P(x)^{1/T}}{Z_T}, \quad Z_T = \sum_x P(x)^{1/T}.$$

The proof is provided in Appendix E.1.

## B.2 Expected Sampling Cost

While Algorithm 1 provides an exact solution for likelihood-free temperature sampling, its practical viability hinges on its computational efficiency. A central concern is the expected number of samples it requires, as each sampler call involves a forward pass through the generative head and autoencoder. Although these forward passes can be executed in parallel during inference, a prohibitively large number of samples would still create a significant computational bottleneck. The following theorem provides a closed-form expression for this expected number of sampler calls, with Corollary 2.1 offering a more interpretable upper bound. The proof is provided in Appendix 2.

**Theorem 2.** *The expected number of calls to the base sampler $S$, denoted $\mathbb{E}[N_{total}]$, required to generate one sample using Algorithm 1 is:*

$$\mathbb{E}[N_{total}] = \frac{n + \mathbb{I}(\alpha > 0) \sum_x P(x)^{1/T-1}}{Z_T}$$

*where $Z_T = \sum_x P(x)^{1/T}$, $n = \lfloor 1/T \rfloor$, $\alpha = 1/T - n$, and $\mathbb{I}(\cdot)$ is the indicator function.*

**Corollary 2.1.** *Let $|\mathcal{X}|$ be the size of sample space. The expected number of sampler calls $\mathbb{E}[N_{total}]$ at temperature $T \in (0,1)$ is bounded by:*

$$\mathbb{E}[N_{total}] \leq \begin{cases} \dfrac{1+n}{Z_T}, & \text{if } 0 < T \leq 0.5 \\ \dfrac{1 + |\mathcal{X}|^{2-1/T}}{Z_T}, & \text{if } 0.5 < T < 1 \end{cases}$$

*where $n = \lfloor 1/T \rfloor$ and $Z_T = \sum_x P(x)^{1/T}$.*

These results highlight that the algorithm's practicality is highly sensitive to the temperature $T$. A potential limitation first emerges for $T \to 1$, as the cost can scale up to the size of sample space $|\mathcal{X}| = |\mathcal{V}|^K$. It is therefore advisable to avoid using temperatures in this high-temperature regime to prevent a potential computational bottleneck. Conversely, at low temperatures, the integer part $n = \lfloor 1/T \rfloor$ becomes large. The algorithm's success requires drawing $n$ identical samples, an event with a vanishingly small probability for a large $n$ that leads to an extremely high rejection rate. A more sample-efficient approximate algorithm is therefore needed to enhance its practical utility.

## B.3 Batch Approximation

The practical limitations of the exact algorithm become most pronounced in the low-temperature regime, where the requirement of drawing $n = \lfloor 1/T \rfloor$ identical samples leads to an extremely high rejection rate that results in poor sample utilization. To address this, we propose an efficient approximate algorithm tailored for low temperatures of the form $T = 1/n$. The key insight is to shift from a single, high-risk trial to a combinatorial search within a large batch of $N$ samples ($N \gg n$). This shift allows a single batch to constitute $\binom{N}{n}$ distinct candidates, which dramatically improves sample utilization and increases the probability of finding a successful match in a single round.

For example, to sample at $T = 0.5(n = 2)$, we might draw a batch of $N = 10$ samples, such as $\{A, C, A, D, B, E, A, F, B, G\}$. Here, sample $A$ appears three times, and sample $B$ appears twice. The algorithm then counts the number of successful $n$-tuple candidates within this batch. For sample $A$, there are $\binom{3}{2} = 3$ successful candidates. For sample $B$, there is only $\binom{2}{2} = 1$ successful candidate. Finally, the output is sampled from the set of valid candidates $\{A, B\}$ according to their weighted probabilities, where $P(A) = 3/4$ and $P(B) = 1/4$. In the rare case that no sample

---

**Algorithm 2** Approximate Temperature Sampling

---

**Input:** A base sampler $S$; Target temperature $T = 1/n$; Batch size $N \gg n$.
**Output:** A sample $x$ approximating the distribution $P_T(x) \propto P(x)^n$.
1: **procedure** APPROXIMATETEMPSAMPLE($S, n, N$)
2:     Draw a batch of $N$ samples $\mathcal{B} = \{x_1, \ldots, x_N\}$ from sampler $S$.
3:     Compute counts $c_x$ for each unique sample $x \in \mathcal{B}$.
4:     **for** $m \leftarrow n$ **down to** 1 **do**                      ▷ Start with the target $n$ and fallback if needed
5:         Initialize candidate set $\mathcal{X}_{\text{cand}} \leftarrow \emptyset$.
6:         Initialize weights list $W \leftarrow \emptyset$.
7:         **for** each unique sample $x$ with count $c_x \geq m$ **do**
8:             Add $x$ to $\mathcal{X}_{\text{cand}}$.
9:             Add weight $w_x = \binom{c_x}{m}$ to $W$.                  ▷ Weight is the number of combinations
10:         **if** $\mathcal{X}_{\text{cand}}$ is not empty **then**
11:             **break**                              ▷ Found a valid candidate set, exit fallback loop
12:     Sample $x_{\text{out}}$ from $\mathcal{X}_{\text{cand}}$ with probabilities proportional to weights in $W$.
13:     **return** $x_{\text{out}}$

---

appears at least $n$ times, the candidate set would be empty. To ensure the algorithm always produces an output, we introduce a fallback mechanism that iteratively reduce the matching requirement from $n$ to $n-1$, $n-2$, $\ldots$, until a non-empty candidate set is found. The detailed process is illustrated in Algorithm 2.

For any finite batch size $N$, the algorithm is biased. This bias arises because the output probability is determined by the ratio of weights calculated within a single stochastic batch, and the expectation of a ratio is generally not equal to the ratio of expectations. However, its key strength is that it is asymptotically unbiased: as the batch size $N$ approaches infinity, the output distribution converges to the true target distribution. We formalize this crucial property in the following theorem.

**Theorem 3.** *Let $P_{alg}(x; N)$ be the probability of sampling $x$ using Algorithm 2 with a batch size of $N$, and let $P_T(x) = P(x)^n/Z_T$ be the true target distribution at temperature $T = 1/n$, where $Z_T = \sum_x P(x)^n$. The algorithm is asymptotically unbiased:*

$$\lim_{N \to \infty} P_{alg}(x; N) = P_T(x).$$

The proof is provided in Appendix E.3. This property of consistency establishes the algorithm as a principled approximation, where the batch size $N$ serves as a practical lever for the trade-off between efficiency and accuracy. Because the algorithm relies solely on a black-box sampling interface, its utility extends naturally beyond the CALM framework to the entire class of implicit language models. This positions it as a universal toolkit for controlled generative modeling in discrete spaces.

## C    RELATED WORK

### C.1    AUTOENCODER

**Latent Generative Modeling.** A prominent paradigm in generative modeling involves a two-stage process: first learning a compressed latent representation of the data, and then training a generative model within that latent space. This approach often begins with a Variational Autoencoder (Kingma & Welling, 2014), which learns a mapping from a high-dimensional data space into a compact, continuous latent space. This principle enables modern architectures, such as latent diffusion models (Rombach et al., 2022; Liu et al., 2023), to efficiently generate high-dimensional data from a continuous latent representation. An alternative path, the Vector Quantized VAE (VQ-VAE, van den Oord et al., 2017), learns a discrete latent space by mapping inputs to a finite, learned codebook. This approach has been foundational to the autoregressive generation of continuous data like images (Razavi et al., 2019; Esser et al., 2021; Ramesh et al., 2021; Sun et al., 2024a) and audio (Dhariwal et al., 2020; Zeghidour et al., 2021; Défossez et al., 2023). Our approach introduces a distinct way by performing a discrete-to-continuous mapping. Driven by the pursuit of efficiency, it fundamentally reduces the number of autoregressive steps required for language generation.

**Text Compression.** Compressing long text into compact vector representations is a foundational concept in sequence modeling. For instance, Recurrent Neural Networks can be viewed as implicitly compressing the entire history of a sequence into a single hidden state vector (Elman, 1990; Hochreiter & Schmidhuber, 1997). In the era of LLMs, this concept has been revitalized, with a focus on prompt compression to improve inference efficiency. For example, Mu et al. (2023) designed a modified attention mechanism to distill prompt information into a few memory tokens. Chevalier et al. (2023); Ge et al. (2024); Gao et al. (2024) further introduced explicit reconstruction objectives to promote high fidelity compression. Recently, Li et al. (2025); Kuratov et al. (2025); Mezentsev & Oseledets (2025) pushed the limits of compression to a ratio up to 1568x, underscoring the inherent sparsity of discrete text representations. However, the primary focus of these methods on prompt compression places a greater emphasis on reconstruction fidelity than on the robustness of the resulting representation. Our work, by contrast, prioritizes the creation of a robust and smooth latent manifold, which is a critical prerequisite for stable downstream generative modeling.

### C.2 Likelihood-Free Language Modeling

**Continuous Autoregressive Generation.** Autoregressive generation over continuous vectors is an emerging research frontier, with notable successes in domains such as image (Tschannen et al., 2023; Li et al., 2024; Shao et al., 2025b; Fan et al., 2025; Team, 2025), video (Chen et al., 2024; Deng et al., 2025), and audio synthesis (Turetzky et al., 2024; Sun et al., 2024b; Ma et al., 2025). GIVT (Tschannen et al., 2023) pioneered this direction by fitting the distribution of the target vector with a Gaussian Mixture Model. However, the expressive power of GIVT is confined to the pre-defined family of Gaussian mixtures, a constraint that limits its ability to capture complex distributions. Li et al. (2024) overcomes this limitation by employing a lightweight diffusion head to model the vector distribution. While being more expressive, this method comes at the cost of inference efficiency due to its iterative sampling process. More recently, Shao et al. (2025b) introduced a general framework based on strictly proper scoring rules. The Energy Transformer was presented as a concrete and powerful instance of this framework, capable of high-quality, single-step generation. Our work adopts the core Energy Transformer framework but introduce several key improvements to the generative head architecture, the energy loss, and the model's input structure, to further enhance its performance and stability for the specific challenges of language modeling.

**Parallel Token Prediction.** The goal of predicting multiple tokens in parallel to overcome the sequential bottleneck of autoregressive models is a long-standing pursuit in sequence modeling. Early efforts in this area were pioneered by non-autoregressive machine translation (Gu et al., 2018; Gu & Kong, 2021; Shao et al., 2021; Shao & Feng, 2022; Huang et al., 2022; Gui et al., 2023), which aims to generate an entire target sentence in a single step. While effective for highly constrained conditional tasks like translation, these methods often struggle with the inherent multi-modality of open-ended language generation. A different line of work uses multi-token prediction to enrich training signals (Gloeckle et al., 2024; Shao et al., 2025a) or provide candidates for speculative decoding (Stern et al., 2018; Leviathan et al., 2023), while the underlying generation remains single-token autoregressive. A more direct approach involves hierarchical modeling, where a global model predicts large semantic chunks, which are then decoded by a local model (Lee et al., 2022; YU et al., 2023; Ho et al., 2024; team et al., 2024; Pagnoni et al., 2025; Neitemeier et al., 2025). For instance, MegaByte (YU et al., 2023) uses a global Transformer to predict blocks of tokens, but still relies on a local autoregressive model to generate tokens sequentially within each block. Conceptually closer to our work, Large Concept Models (team et al., 2024) also adopt a hierarchical structure, where their global model autoregressively predicts continuous sentence embeddings. However, this approach faces several challenges that our CALM framework is designed to address: its SONAR autoencoder (Duquenne et al., 2023) is computationally heavy and fragile, and its reliance on a diffusion-based generative process introduces a iterative inference bottleneck. Finally, another paradigm for parallel generation is diffusion models for text, which iteratively refine a sequence of tokens from noise, either at the full sentence (Austin et al., 2021; Li et al., 2022; Lou et al., 2024) or block level (Han et al., 2023; Arriola et al., 2025). These models, which currently operate in the challenging discrete token space, could potentially benefit from the robust continuous space our autoencoder provides.

### C.3 LIKELIHOOD-FREE LM EVALUATION

**LM metrics.** The evaluation of language models is split into two distinct paradigms, which reflects the separation between assessing the quality of generated output and the fidelity of the learned distribution. On one hand, likelihood-based metrics, such as Perplexity, offer a principled way to evaluate the learned distribution, but they are limited to models where likelihoods are tractable. On the other hand, a diverse family of sample-based metrics focuses on the generated output. Classic methods like BLEU (Papineni et al., 2002) and ROUGE (Lin, 2004) assess the quality of generated text by comparing it to reference outputs. More recent approaches such as MAUVE (Pillutla et al., 2021) or LLM-as-a-judge (Zheng et al., 2023) allows for reference-free evaluation, but they rely on heuristics or black-box models and lack the formal guarantees of scoring rules. Our proposed metric, BrierLM, is designed to bridge this gap by combining the advantages of both paradigms: it operates exclusively on model samples, yet as a strictly proper scoring rule, it offers a faithful assessment of the model's predictive quality, akin to perplexity.

**Brier Score.** The Brier Score was originally proposed by Brier (1950) for the evaluation of probabilistic weather forecasts. It is a classic example of a strictly proper scoring rules, theoretically guaranteeing that a model is incentivized to report its true belief to achieve the optimal score (Gneiting & Raftery, 2007). Consequently, it has been widely adopted in classification tasks, primarily for evaluating the quality of probabilistic forecasts (Sanders, 1963; fer, 2009; Hui & Belkin, 2021) and assessing model calibration (Lakshminarayanan et al., 2017; Ovadia et al., 2019; Gruber & Buettner, 2022). The innovation of our work is twofold: first, we introduce a method to unbiasedly estimate the Brier score in a likelihood-free manner; second, we generalize its application from a metric for simple classification tasks to one capable of assessing language modeling capabilities.

### C.4 LIKELIHOOD-FREE TEMPERATURE SAMPLING

**Bernoulli Factory.** The temperature sampling problem is conceptually related to the classic problem of the Bernoulli Factory (Keane & O'Brien, 1994; Occil, 2020), which addresses the challenge of simulating a new coin with a success probability of $f(p)$ given only a coin with an unknown success probability $p$. This mirrors our challenge of achieving a target probability proportional to $P(x)^{1/T}$ using only a base sampler for the implicit distribution $P(x)$. A key distinction is that the Bernoulli Factory problem assumes a binary outcome, whereas we operate over a large discrete sample space. Our two-stage algorithm elegantly bridges this gap. The first stage isolates a single candidate $x^*$ and reduces the problem to a binary one, and the second stage directly applies an existing Bernoulli Factory algorithm (Mendo, 2019) to construct an event with probability $P(x^*)^{\alpha}$.

**Controlled Generation.** While many generative models lack the explicit probabilistic controls for temperature sampling, they have developed alternative strategies to navigate the trade-off between sample quality and diversity. For instance, VAEs and normalizing flows (Kingma & Welling, 2014; Rezende & Mohamed, 2015) often achieve this by adjusting the variance of their prior latent distribution (Kingma & Dhariwal, 2018). In Generative Adversarial Networks (Goodfellow et al., 2014), the truncation trick restricts sampling to a high-density region of the latent space (Brock et al., 2019). Similarly, diffusion models can control stochasticity by altering the noise variance during the reverse sampling process (Song et al., 2021). These techniques, however, are fundamentally heuristic, as it is generally intractable to characterize the shape of the modified output distribution, and they all require white-box access to model internals like the latent space. Our work, in contrast, proposes a universal, black-box algorithm for temperature sampling from implicit models over discrete spaces, offering a provably exact method for this broad class of models.

## D SUPPLEMENTARY EXPERIMENTS

### D.1 EFFECT OF AUTOENCODER

In this section, we study the effect of the autoencoder's design choices on the final performance of the CALM framework. The autoencoder is a critical component, as it defines the latent space in which the continuous language model operates. To isolate its effects, we hold the downstream language model fixed across all experiments in this section: an Energy Transformer with a hidden size of 768, 12 hidden layers, 16 attention heads, an FFN intermediate size of 2048, and a generative

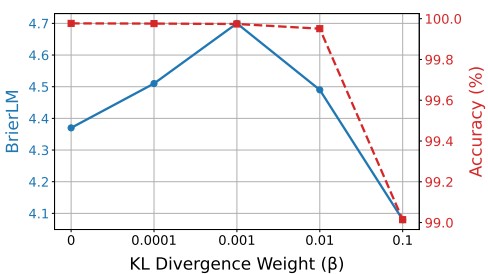 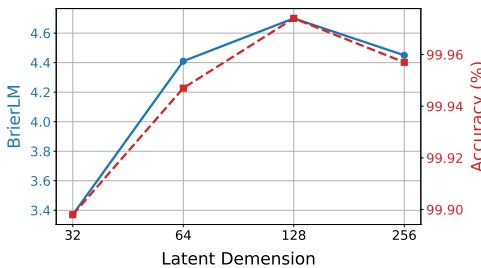

Figure 6: Effect of the KL divergence weight on the autoencoder's reconstruction accuracy and the downstream BrierLM score.

Figure 7: Effect of the latent demension on the autoencoder's reconstruction accuracy and the downstream BrierLM score.

head with 3 MLP blocks. Each model configuration is trained for 50,000 steps. Unless otherwise specified, the autoencoder uses the default parameters as described in Section 6.1. We begin with a comprehensive ablation study to validate the contribution of each proposed technique, followed by a detailed analysis of the effect of several key hyperparameters.

We first validate the design choices for the autoencoder, with results detailed in Table 2. While a standard, reconstruction-only autoencoder provides a reasonable baseline, naively incorporating a variational objective leads to a significant drop in performance. This degradation is traced to a severe instance of posterior collapse, where we found that 71 of the 128 latent dimensions had collapsed to the standard normal prior. The introduction of the KL clipping strategy proves to be the crucial remedy, which effectively prevents dimensional collapse and leads to a notable performance improvement. Furthermore, applying dropout regularization to both the input tokens and the latent vector yields considerable, orthogonal performance benefits, confirming that each technique contributes uniquely to shaping a high-fidelity and robust latent space.

Table 2: Ablation study of the autoencoder's regularization techniques. Performance is measured by BrierLM on the downstream language modeling task.

| $\mathcal{L}_{\text{KL}}$ | $\mathcal{L}_{\text{KL}}^{clip}$ | DropToken | DropLatent | BrierLM |
|:---:|:---:|:---:|:---:|:---:|
| | | | | 3.99 |
| ✓ | | | | 3.48 |
| | ✓ | | | 4.13 |
| | ✓ | ✓ | | 4.55 |
| | ✓ | | ✓ | 4.46 |
| | ✓ | ✓ | ✓ | 4.70 |

**KL weight.** We next examine the model's sensitivity to the KL divergence weight, $\beta$, which governs the trade-off between reconstruction fidelity and latent space regularization. We varied $\beta$ across several orders of magnitude and present the results in Figure 6. Starting from a baseline with no KL regularization ($\beta = 0$), we observe that introducing a small amount of variational regularization significantly improves the final BrierLM score, which confirms our hypothesis: a moderate regularization effectively smooths the latent manifold, making it more learnable for the Energy Transformer, while leaving reconstruction accuracy almost unaffected. However, this trend reverses as the regularization becomes overly aggressive. At $\beta = 0.1$, the BrierLM score drops sharply, a decline directly linked to the autoencoder's compromised reconstruction fidelity, which falls to $\sim$99%. Based on these findings, we selected $\beta = 0.001$ to train our autoencoder.

**Latent Demension.** We next examine the influence of the latent dimension, $l$, which functions as the information bottleneck of the autoencoder. We evaluated latent dimensions of 32, 64, 128, and 256, and respectively scaled the dropout rate to 0.05, 0.1, 0.15, and 0.2. The results are illustrated in Figure 7. As seen, the reconstruction accuracy remains consistently high across all configurations, but the downstream performance varies, peaking at $l = 128$. This suggests a trade-off in selecting the optimal dimension. A latent space that is too small, such as with $l = 32$, forces the autoencoder

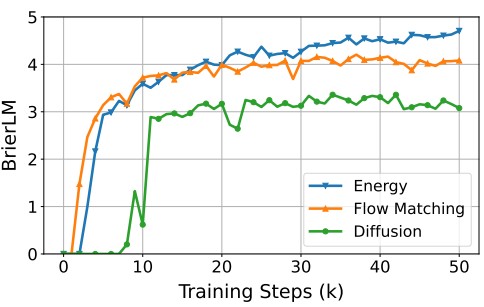

Figure 8: BrierLM scores during training for different generative heads.

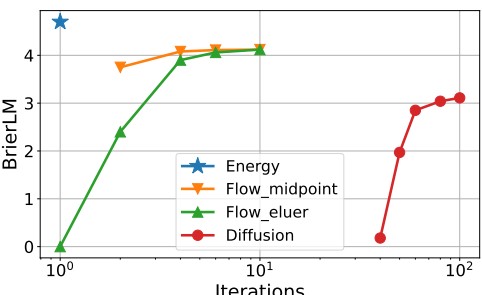

Figure 9: Effect of sampling steps on the generation quality of diffusion and flow matching.

to learn an overly compact and brittle representation. Conversely, a large latent dimension may lead the autoencoder to encode noisy or irrelevant features from the input tokens. This forces the Energy Transformer to expend its finite capacity modeling this noise, making it more challenging to discern the underlying data manifold. A dimension of $l = 128$ thus appears to strike an optimal balance, providing sufficient capacity for a robust representation while maintaining a structured and learnable latent space for the downstream generative model.

**Scale.** Finally, we examine the impact of scaling the autoencoder. We explored several axes of scaling: doubling the number of layers in both the encoder and decoder to 4, doubling the hidden dimension to 1024, and expanding the training dataset to 100B tokens. Interestingly, none of these scaling efforts resulted in a significant improvement in the final BrierLM score. This finding suggests that the autoencoder's task is inherently simple and does not benefit from the aggressive scaling. A lightweight architecture, trained on a relatively modest amount of data, is sufficient to learn the high-fidelity and robust representation required for our framework. This is a desirable property, as it allows the autoencoder to be a computationally negligible component of the overall system.

## D.2 EFFECT OF MODEL ARCHITECTURE

In this section, we conduct ablation studies on the CALM model architecture to investigate the impact of different design choices on model performance. Unless otherwise specified, all experiments are conducted using a base configuration with a hidden size of 768, 12 hidden layers, 16 attention heads, and an FFN intermediate size of 2048. The generative head consists of 3 MLP blocks, and all models are trained for 50,000 steps.

**Diffusion and Flow Matching.** Since the generative head can be any continuous generative model, we also evaluate two prominent choices as alternatives: diffusion (Ho et al., 2020) and flow matching (Lipman et al., 2023). For these experiments, we adopt an architecture consistent with the diffusion head used in Li et al. (2024). To ensure a fair comparison, we replicate the input hidden state $N = 8$ times for both models, mirroring the multi-sample approach used for our energy loss and promoting stable learning. During inference, we use 100 iterative steps by default. For the Flow Matching model, we use a midpoint sampler. Figure 8 compares the performance of the diffusion, flow matching, and energy-based generative heads. The results show that both flow matching and our energy-based head outperform the diffusion model, exhibiting a noticeable performance gap. Between the two, flow matching exhibits faster initial convergence, whereas our energy-based head reaches a higher performance ceiling.

Figure 9 further compares the models' performance across different numbers of inference iterations. For the flow matching model, we tested both the Euler and midpoint samplers. As shown, the diffusion model requires a large number of iterations to generate valid results. In contrast, the flow matching model is significantly more efficient; the midpoint sampler, in particular, achieves decent quality in just 2 steps and reaches its near-optimal performance within 4 steps. Our energy-based generative head achieves the best of both worlds: it delivers superior performance while completely eliminating the need for iterative decoding, making it a compelling choice for the CALM framework.

**Energy Loss.** We now analyze the impact of the energy loss formulation on model performance. Our energy loss (Equation 10) involves two sampling hyperparameters: the number of model-generated

samples, $N$, and the number of target samples, $M$. Larger values for $N$ and $M$ provide a better estimation of the true energy score, but also increase the computational cost. Our default configuration is $N = 8$ and $M = 100$. Table 3 shows the results of varying $N$ and $M$, which reveal a clear trade-off between performance and computational cost. As expected, increasing the number of samples consistently improves the BrierLM score, but this comes at a nearly linear increase in training cost. Our default setting of $N = 8$ and $M = 100$ is thus justified as a balanced configuration, leveraging a moderately sized $N$ for a robust gradient signal and a large $M$ to stabilize training.

Table 3: Effect of model samples $N$ and target samples $M$ on model performance and training cost.

|  | Varying $N$ (fixed $M = 100$) | | | | Varying $M$ (fixed $N = 8$) | | | |
|---|---|---|---|---|---|---|---|---|
|  | $N = 2$ | $N = 4$ | $N = 8$ | $N = 12$ | $M = 1$ | $M = 16$ | $M = 100$ | $M = 200$ |
| BrierLM | 4.37 | 4.53 | 4.70 | 4.72 | 4.50 | 4.56 | 4.70 | 4.67 |
| Relative Cost | 0.82× | 0.91× | 1.0× | 1.13× | 0.92× | 0.94× | 1.0× | 1.07× |

We also investigate the effect of the exponent $\alpha$ in the energy score (Equation 9), which is guaranteed to be strictly proper for any $\alpha \in (0, 2)$. As shown in Table 4, our empirical results align with this theoretical property. We observe that training fails for $\alpha < 1$ (e.g., $\alpha = 0.75$), a phenomenon previously analyzed by Shao et al. (2025b) and attributed to gradient explosion issues. For values of $\alpha$ within the range of $[1, 2)$, the model achieves decent performance, with the best empirical results obtained at our default setting of $\alpha = 1$. The model's BrierLM score drops to 0 at $\alpha = 2$. This is expected, as the energy score is only proper but not strictly proper when $\alpha = 2$. Consequently, the energy loss can no longer guide the model to uniquely match the true data distribution, leading to a collapse in modeling capability.

Table 4: Effect of the exponent $\alpha$ in the energy score.

| $\alpha$ | 0.75 | 1 | 1.25 | 1.5 | 1.75 | 2 |
|---|---|---|---|---|---|---|
| BrierLM | Fail | 4.70 | 4.42 | 4.46 | 4.30 | 0 |

**Model Input.** A critical design choice in our CALM framework is the input representation fed into the Transformer backbone at each autoregressive step. We evaluate three distinct input schemes: (1) Discrete input, which first decodes the previously generated vector $\mathbf{z}_{i-1}$ into $K$ discrete tokens, then passes them through an embedding layer and an input compression MLP to form the next step input; (2) Continuous input, a more direct alternative where the vector $\mathbf{z}_{i-1}$ is directly projected to the Transformer's hidden dimension via a single linear layer; (3) Combined input, which fuses the representations from the discrete and continuous methods through element-wise addition.

Table 5: Effect of model input on language modeling performance. Performance is evaluated using Brier-n scores and the composite BrierLM. Higher scores are better.

| Model Input | Brier-1 | Brier-2 | Brier-3 | Brier-4 | BrierLM |
|---|---|---|---|---|---|
| Discrete | 21.81 | 6.88 | 2.59 | 1.25 | 4.70 |
| Continuous | 17.43 | 5.04 | 1.74 | 0.73 | 3.25 |
| Both | 21.17 | 6.49 | 2.44 | 1.12 | 4.40 |

As summarized in Table 5, the results clearly favor the discrete input strategy. The combined input offers no advantage and slightly degrades performance, while the purely continuous input leads to a substantial performance drop. This confirms our hypothesis: although the continuous vector theoretically contains all the information of its corresponding discrete tokens, its highly compact and brittle nature makes it challenging for the model to unpack the underlying semantic information. Grounding the autoregressive process in the discrete token space provides a more structured and stable input signal, which is therefore critical for achieving optimal performance.

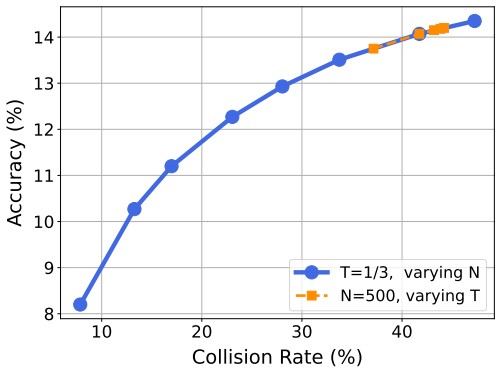

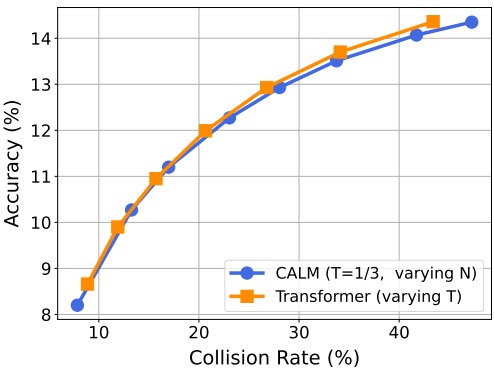

Figure 10: The accuracy-diversity trade-off in CALM as a function of batch size N and temperature T.

Figure 11: Comparison of the temperature sampling performance between CALM and the baseline Transformer.

### D.3 TEMPERATURE SAMPLING

In this section, we conduct a fine-grained analysis to characterize the practical behavior of our approximate temperature sampling algorithm (Algorithm 2), i.e., how the algorithm navigates the trade-off between predictive accuracy and generative diversity. To quantify them, we decompose the Brier score estimator, $\text{Brier}(P, y) \approx \mathbb{I}\{x_1 = y\} + \mathbb{I}\{x_2 = y\} - \mathbb{I}\{x_1 = x_2\}$, into two metrics:

- **Accuracy** $\mathbb{E}[\mathbb{I}\{x = y\}]$: This metric measures the probability that a single sample drawn from the model matches the ground truth. It directly reflects the model's accuracy.

- **Collision Rate** $\mathbb{E}[\mathbb{I}\{x_1 = x_2\}]$: This metric measures the collision probability that two independent samples drawn from the model are identical. It serves as an inverse proxy for diversity, where a higher collision rate indicates that the output distribution is less diverse.

Consistent with our primary BrierLM metric, we report both accuracy and collision rate as the geometric mean of scores over n-grams from n=1 to 4. We first investigate how the two key hyperparameters of our algorithm—temperature $T$ and batch size $N$—influence these metrics. Specifically, we conduct two sets of experiments: for a fixed temperature $T = 1/3$, we vary the batch size $N \in \{1, 10, 20, 50, 100, 200, 500, 1000\}$; for a fixed batch size $N = 500$, we vary the the temperature $T \in \{1/2, 1/3, 1/4, 1/5, 1/6\}$.

As shown in Figure 10, both increasing the batch size $N$ and decreasing the temperature $T$ sharpen the output distribution, achieving higher accuracy at the cost of reduced diversity (i.e., a higher collision rate). A key observation, however, is the dominant role of the batch size $N$, which covers a substantially greater range of this trade-off than the temperature $T$. Intuitively, a larger batch provides a clearer statistical picture of the true distribution, making it easier to indentify high-probability candidates and confidently output them. In contrast, the effectiveness of temperature $T$ is capped by the information available within the finite batch. Thus, while temperature serves its conventional purpose, these empirical results suggest that the batch size $N$ is a more effective tool for navigating the accuracy-diversity frontier in our likelihood-free framework.

Finally, we compare the behavior of our sampling algorithm against that of a traditional Transformer. We ensure a fair comparison by selecting model checkpoints with nearly identical BrierLM scores. For CALM, we fix the temperature at $T = 1/3$ while varying the batch size $N \in \{1, 10, 20, 50, 100, 200, 500, 1000\}$; for the Transformer baseline, we adjust its softmax temperature across $T \in \{1, 0.9, \ldots, 0.4\}$.

The results, plotted in Figure 11, are compelling: the accuracy-diversity trajectory traced by tuning $N$ in CALM is nearly identical to the one produced by tuning $T$ in the traditional Transformer. This alignment shows that we can accurately replicate the generative behavior of traditional models across a wide spectrum of temperatures. For instance, matching $T = 0.6$ requires a batch size of approximately $N = 100$, while simulating a lower temperature of $T = 0.5$ necessitates a larger

batch of $N = 200$. This suggests a clear and predictable trade-off: the ability to simulate lower-temperature, higher-fidelity generation comes at the cost of an increased number of samples.

# E    PROOF

## E.1    PROOF OF THEOREM 1

**Theorem 1.** *For an implicit discrete distribution $P(x)$ with sampler $S$ and a temperature $T \in (0, 1)$, Algorithm 1 generates samples distributed as:*

$$P_T(x) = \frac{P(x)^{1/T}}{Z_T}, \quad Z_T = \sum_x P(x)^{1/T}.$$

*Proof.* Algorithm 1 implements a rejection sampling scheme where sample $x$ is accepted with probability $P_{\text{accept}}(x)$. The proof proceeds by showing that the acceptance probability $P_{\text{accept}}(x) = P(x)^{1/T}$, so the rejection sampling procedure yields a normalized sample distribution as:

$$P_T(x) = \frac{P(x)^{1/T}}{\sum_x P(x)^{1/T}}. \tag{16}$$

The inverse temperature is decomposed as $1/T = n + \alpha$, where $n = \lfloor 1/T \rfloor$ is the integer part and $\alpha \in [0, 1)$ is the fractional part. The acceptance probability is the product of the success probabilities of the two corresponding stages.

**Stage 1 (Integer Part):** For the algorithm to proceed to stage 2 with a candidate sample $x$, it must first draw $x$ for $n$ consecutive times in stage 1. As each draw is independent with probability $P(x)$, the probability of passing stage 1 with candidate $x$ is $P(x)^n$.

**Stage 2 (Fractional Part):** Let $p = P(x)$. The probability of acceptance in stage 2 is the sum of probabilities of accepting at each iteration $i \geq 1$:

$$
\begin{aligned}
P_{\text{stage2}} &= P(\text{accept at } i = 1) + P(\text{pass } i = 1, \text{accept at } i = 2) + \dots \\
&= p + (1 - p)\left(1 - \frac{\alpha}{1}\right)p + (1 - p)^2 \left(1 - \frac{\alpha}{1}\right)\left(1 - \frac{\alpha}{2}\right)p + \dots \\
&= p \sum_{k=0}^{\infty} (1 - p)^k \prod_{j=1}^{k} \left(1 - \frac{\alpha}{j}\right) \\
&= p \sum_{k=0}^{\infty} (p - 1)^k \binom{\alpha - 1}{k} \\
&= p \cdot (p - 1 + 1)^{\alpha - 1} \\
&= p^{\alpha}.
\end{aligned}
\tag{17}
$$

The fifth equation is the application of the generalized binomial theorem.

**Total Probability:** The total probability of accepting a sample $x$ in a single trial is the product of the probabilities from the two stages:

$$P_{\text{accept}}(x) = P(x)^n \cdot P(x)^{\alpha} = P(x)^{1/T}, \tag{18}$$

which completes the proof. $\qquad\square$

### E.2 PROOF OF THEOREM 2

**Theorem 2.** *The expected number of calls to the base sampler $S$, denoted $\mathbb{E}[N_{total}]$, required to generate one sample using Algorithm 1 is:*

$$\mathbb{E}[N_{total}] = \frac{n + \mathbb{I}(\alpha > 0)\sum_x P(x)^{1/T-1}}{Z_T}$$

*where $Z_T = \sum_x P(x)^{1/T}$, $n = \lfloor 1/T \rfloor$, $\alpha = 1/T - n$, and $\mathbb{I}(\cdot)$ is the indicator function.*

*Proof.* The algorithm is a series of independent trials, which continues until one trial is successful. This follows a geometric distribution. The total expected number of samples is the ratio of the expected number of samples per trial, $\mathbb{E}[N_{trial}]$, to the probability of a trial's success, $P(\text{success})$:

$$\mathbb{E}[N_{total}] = \frac{\mathbb{E}[N_{trial}]}{P(\text{success})}. \tag{19}$$

**Denominator (Success Probability):** A trial is successful if any sample $x$ is accepted. The total success probability is the sum of acceptance probabilities over all possible outcomes:

$$P(\text{success}) = \sum_x P_{\text{accept}}(x) = \sum_x P(x)^{1/T} = Z_T. \tag{20}$$

**Numerator (Expected Calls per Trial):** Let $N_{trial}$ be the number of sampler calls in a single trial. A trial always involves $N_1$ calls for stage 1 and may involve $N_2$ calls for stage 2. The number of calls in Stage 1 is fixed at $N_1 = n$. Thus, $\mathbb{E}[N_1] = n$.

If $\alpha = 0$, stage 2 is never performed, so $\mathbb{E}[N_2] = 0$. If $\alpha > 0$, stage 2 is only executed if stage 1 succeeds with some candidate $x$. Let $\mathcal{E}_x$ be this event, which occurs with probability $P(\mathcal{E}_x) = P(x)^n$. We have:

$$\mathbb{E}[N_2] = \sum_x P(\mathcal{E}_x) \cdot \mathbb{E}[N_2|\mathcal{E}_x]. \tag{21}$$

The conditional expectation $\mathbb{E}[N_2|\mathcal{E}_x]$ is the expected number of draws in stage 2 given candidate $x$. Using the formula $\mathbb{E}[X] = \sum_{k=1}^{\infty} P(X \geq k)$, where $X$ is the number of draws in stage 2:

$$\mathbb{E}[N_2|\mathcal{E}_x] = \sum_{k=1}^{\infty} P(\text{stage 2 requires at least } k \text{ draws})$$
$$= \sum_{k=0}^{\infty}(1 - P(x))^k \prod_{j=1}^{k}\left(1 - \frac{\alpha}{j}\right). \tag{22}$$

This is the same sum we evaluated in Equation 17, which equals $P(x)^{\alpha-1}$. Therefore, if $\alpha > 0$:

$$\mathbb{E}[N_2] = \sum_x P(x)^n \cdot P(x)^{\alpha-1} = \sum_x P(x)^{n+\alpha-1} = \sum_x P(x)^{1/T-1}. \tag{23}$$

Combining the two cases, the total expected number of calls per trial is:

$$\mathbb{E}[N_{\text{trial}}] = \mathbb{E}[N_1] + \mathbb{E}[N_2] = n + \mathbb{I}(\alpha > 0)\sum_x P(x)^{1/T-1}. \tag{24}$$

Combining the numerator and denominator gives the final result:

$$\mathbb{E}[N_{\text{total}}] = \frac{n + \mathbb{I}(\alpha > 0)\sum_x P(x)^{1/T-1}}{Z_T}. \tag{25}$$

$\square$

**Corollary 2.1.** *Let $|\mathcal{X}|$ be the size of sample space. The expected number of sampler calls $\mathbb{E}[N_{total}]$ at temperature $T \in (0, 1)$ is bounded by:*

$$\mathbb{E}[N_{total}] \leq \begin{cases} \dfrac{1+n}{Z_T}, & \text{if } 0 < T \leq 0.5 \\ \dfrac{1 + |\mathcal{X}|^{2-1/T}}{Z_T}, & \text{if } 0.5 < T < 1 \end{cases}$$

*where $n = \lfloor 1/T \rfloor$ and $Z_T = \sum_x P(x)^{1/T}$.*

*Proof.* The proof is divided into two cases based on the temperature range. We start from the general formula for the expected cost from Theorem 2:

$$\mathbb{E}[N_{\text{total}}] = \frac{n + \mathbb{I}(\alpha > 0) \sum_x P(x)^{1/T-1}}{Z_T}. \tag{26}$$

**Case 1: Low-Temperature Regime ($0 < T \leq 0.5$)**

In this range, the exponent $1/T - 1 \geq 1$. Since $P(x) \in [0, 1]$, for any exponent $\beta \geq 1$, we have $P(x)^\beta \leq P(x)$. Thus, by summing over the entire sample space $\mathcal{X}$:

$$\sum_{x \in \mathcal{X}} P(x)^{1/T-1} \leq \sum_{x \in \mathcal{X}} P(x) = 1. \tag{27}$$

The numerator of the cost formula is therefore bounded by $n + 1$:

$$n + \mathbb{I}(\alpha > 0) \sum_{x \in \mathcal{X}} P(x)^{1/T-1} \leq n + \mathbb{I}(\alpha > 0) \cdot 1 \leq n + 1. \tag{28}$$

This establishes the bound for the low-temperature regime.

**Case 2: High-Temperature Regime ($0.5 < T < 1$)**

In this range, the exponent $\beta = 1/T - 1$ is in the interval $(0, 1)$. For such an exponent, the function $f(p) = p^\beta$ is concave. By Jensen's inequality, the sum $\sum_{x \in \mathcal{X}} P(x)^\beta$ is maximized when $P(x)$ is a uniform distribution over the sample space, i.e., $P(x) = 1/K$ for all $x \in \mathcal{X}$. The bound is:

$$\sum_{x \in \mathcal{X}} P(x)^{1/T-1} \leq \sum_{x \in \mathcal{X}} \left(\frac{1}{K}\right)^{1/T-1} = K \cdot \left(\frac{1}{K}\right)^{1/T-1} = K^{2-1/T}. \tag{29}$$

Substituting this into the cost formula gives the bound for the high-temperature regime. This completes the proof. $\qquad\square$

### E.3 PROOF OF THEOREM 3

**Theorem 3.** *Let $P_{alg}(x; N)$ be the probability of sampling $x$ using Algorithm 2 with a batch size of $N$, and let $P_T(x) = P(x)^n/Z_T$ be the true target distribution at temperature $T = 1/n$, where $Z_T = \sum_x P(x)^n$. The algorithm is asymptotically unbiased:*

$$\lim_{N \to \infty} P_{alg}(x; N) = P_T(x).$$

*Proof.* Let $\mathcal{B} = \{x_1, \ldots, x_N\}$ be a batch of $N$ samples drawn i.i.d. from the base distribution $P(x)$. For any sample $x \in \mathcal{X}$, let $C_x$ be the random variable for the count of $x$ in $\mathcal{B}$. The weight assigned to $x$ is $W_x = \binom{C_x}{n}$. Let $X_N$ be the random variable for the probability of sampling $x$ from $\mathcal{B}$:

$$X_N(x) = \frac{W_x}{\sum_{z \in \mathcal{X}} W_z} = \frac{\binom{C_x}{n}}{\sum_{z \in \mathcal{X}} \binom{C_z}{n}}. \tag{30}$$

The overall probability we seek to analyze is the expectation of this random variable: $P_{alg}(x; N) = \mathbb{E}[X_N]$. Our goal is to show that $\lim_{N \to \infty} \mathbb{E}[X_N] = P_T(x)$. The proof proceeds in two main steps: first, we show that the random variable $X_N$ converges in probability to $P_T(x)$; second, we use the Bounded Convergence Theorem to show that this implies the convergence of its expectation.

**1. Convergence in Probability.** By the Weak Law of Large Numbers, the proportion of occurrences of any sample $x$ converges in probability to its true probability $P(x)$:

$$\frac{C_x}{N} \xrightarrow{p} P(x) \quad \text{as } N \to \infty. \tag{31}$$

The weight $W_x$ can be written as a polynomial in $C_x$, $W_x = \frac{1}{n!} C_x (C_x - 1) \ldots (C_x - n + 1)$. We normalize this by $N^n$:

$$\frac{W_x}{N^n} = \frac{1}{n!} \left( \frac{C_x}{N} \right) \left( \frac{C_x - 1}{N} \right) \ldots \left( \frac{C_x - n + 1}{N} \right). \tag{32}$$

Since $\frac{C_x}{N} \xrightarrow{p} P(x)$ and $\frac{k}{N} \to 0$ for any constant $k$, each term in the product converges to $P(x)$. By the Continuous Mapping Theorem, the entire expression converges in probability:

$$\frac{W_x}{N^n} \xrightarrow{p} \frac{1}{n!} P(x)^n. \tag{33}$$

Now, we analyze the random variable $X_N$ by dividing its numerator and denominator by $N^n$:

$$X_N = \frac{W_x/N^n}{\sum_{z \in \mathcal{X}} (W_z/N^n)}. \tag{34}$$

Applying the Continuous Mapping Theorem again for the ratio, we show that the random variable $X_N$ converges in probability to $P_T(x)$:

$$X_N \xrightarrow{p} \frac{\frac{1}{n!} P(x)^n}{\sum_{z \in \mathcal{X}} \frac{1}{n!} P(z)^n} = \frac{P(x)^n}{\sum_{z \in \mathcal{X}} P(z)^n} = P_T(x). \tag{35}$$

**2. Convergence of Expectation.** We have established that the random variable $X_N$ converges in probability to $P_T(x)$. Besides, we have that $X_N$ is inherently bounded:

$$0 \leq X_N = \frac{W_x}{\sum_{z \in \mathcal{X}} W_z} \leq 1. \tag{36}$$

We can now invoke the Bounded Convergence Theorem, which states that if a sequence of random variables $X_N$ converges in probability to $X$, and $|X_N| \leq M$ for all $N$ for some constant $M$, then $\lim_{N \to \infty} \mathbb{E}[X_N] = \mathbb{E}[\lim_{N \to \infty} X_N]$. Applying this theorem to our case:

$$\lim_{N \to \infty} P_{alg}(x; N) = \lim_{N \to \infty} \mathbb{E}[X_N] = \mathbb{E}\left[ \lim_{N \to \infty} X_N \right] = P_T(x). \tag{37}$$

This completes the proof that the algorithm is asymptotically unbiased.

$\square$

# F LLM USAGE

We use LLMs as a writing-assistant tool. The role of LLMs is improving the clarity, readability, and grammatical correctness of the text. This included tasks such as refining sentence structure, correcting spelling and grammar, and suggesting alternative phrasing to enhance the manuscript's quality. The authors have carefully reviewed and edited all text, taking full responsibility for content presented in this paper.

