# OpenReview forum: "Continuous Autoregressive Language Models"
_ICLR.cc/2026/Conference — Submitted to ICLR 2026_

### Official Review · Reviewer_FBcT · 2025-10-30

**Soundness:** 3
**Presentation:** 3
**Contribution:** 2
**Rating:** 4
**Confidence:** 4

**Summary:**

The authors introduce an architecture change to transformers, that allow to run the transformer backbone on multiple tokens. I.e. a compression module is introduced that merges the embeddings of k tokens into 1, after standard transformer, the hidden state is decoded thru a series of noise inductions and a variational AE that directly outputs the k next tokens. Like that, it contributes towards more efficient systems as well as an interesting shift in LLM modeling objectives.

**Strengths:**

- in principle interesting idea, in a relevant field. the compression aspect as well as shift to monte carlo based loss instead of CE.
- good writing, easy to follow
- plausible construction towards the experiments

**Weaknesses:**

Major weakness that leads to my reject is the lack of comparisons and failed demonstration of language modeling practicality. Yes you demonstrated somewhat stable scaling law when comparing Brier-Score performance, but:

W1 there is no actual language modeling performance shown anywhere, such as exact-match decoding on token on a corpus. it is not clear that the brier score is useful for language modeling and a simple correlation plot to CE in appendix is not enough. the experiment section in main paper imho is pretty delusive/ insufficient.

W2 while cited, there is no comparison to other multi-token methods like Gloeckle. hierarchical approaches, that also compress the input, have also been there.

W3 if i were to abstract W1 and W2 and 'just focus on the 'paradigm shift'', there are further ablations missing:
- 3.1 why / how many of those stochastic MLP's do i need? just to get a 'less brittle' distribution? theoretically they should not be required?
- 3.2 why is the compressor deterministic?
- 3.3 can one remain in a fully distributional transformer, i.e. re-inject the 'continuous prediction' for otherwise it's really just a multi-token predictor?
- 3.4 i feel like the paper failed to pitch the benefit for the continuous/ likelihood-freeness


Minors
- the 'energy based head' is really just a few noisy linear layers followed by the anyway distributional  VAE - it feels a bit overselling.

**Questions:**

plans to address the weaknesses

---

### Official Review · Reviewer_fiNd · 2025-10-31

**Soundness:** 3
**Presentation:** 3
**Contribution:** 3
**Rating:** 4
**Confidence:** 4

**Summary:**

This paper proposes a novel method to train language models by predicting vectors that represent a window of k tokens. This allows the model to generate k tokens per execution, therefore reducing the compute needed per token. The authors argue that individual tokens carry very minimal amount of information and per token log-likelihood estimates is a limiting factor to achieving better information density. Therefore, they propose to build an auto-encoder to compress sets of contiguous tokens to vector representations and reconstruct them back to token space. The main language modeling task is therefore to predict these vectors in an auto-regressive manner. The rest of the paper describes the challenges of using usual log likelihood estimates for training such a model along with proposed likelihood free estimator. The authors also propose a likelihood free evaluation and temperature sampling method for decoding.

**Strengths:**

1. A new vision of looking into LLM training.
2. Thorough process to define new loss functions, metrics and decoding mechanism for the proposed LM.

**Weaknesses:**

1. Comparisons with existing multi-token prediction methods
2. Accuracy of generative benchmarks
3. Although not explicitly but some papers like 1. Medusa: https://arxiv.org/abs/2401.10774, 2: Your LLM knows the future, https://arxiv.org/pdf/2507.11851 encode this set of token behavior in the model embedding in its current form. So a section explaining this correlation can be useful.

**Questions:**

1. How does the training loop look like?
2. What would happen if I train the architecture (fig:2) end to end on general CE-Loss?

---

### Official Review · Reviewer_yTzS · 2025-11-02

**Soundness:** 2
**Presentation:** 3
**Contribution:** 3
**Rating:** 4
**Confidence:** 3

**Summary:**

This paper introduces Continuous Autoregressive Language Models (CALM), which replace discrete token-by-token prediction with continuous next-vector prediction.
Compressing multiple tokens into a single vector results in efficiency in generation.
To enable CALM, the paper introducies a likelihood-free, continuous generative framework.

**Strengths:**

- CALM reduces both train and inference FLOPs compared to a vanilla transformer
- the generation quality also seems to be better (according to Brier Score)

**Weaknesses:**

- my main concern is limited evaluations:
    - How does CALM perform on standard LM evals like HellaSwag, PIQA etc.?
    - More importantly, i'm curious about in-context recall abilities of CALM. It has been observed that many efficient architectures match vanilla transformer in perplexity, simple LM evals etc., but they lack the ability to recall specific tokens from the past. How does CALM perform on tasks from the EVAPORATE suite: https://huggingface.co/collections/hazyresearch/evaporate-suite ? or needle-in-haystack tasks from RULER benchmark? or even on toy datasets like MQAR [1, 2]?

- perhaps a few comparison with baselines is also missing I think (authors need not run all baselines if they can be differentiated conceptually :) )
    - How is CALM different from Large Concept Models [3]?
    - should Block Transformer or MegaByte be a baseline? even though conceptually they are different from CALM, but since the end-goal is same as CALM i.e. achieving lower FLOPs during inference and training, should these be compared?

[1]: Zoology: Measuring and Improving Recall in Efficient Language Models

[2]: Simple linear attention language models balance the recall-throughput tradeoff

[3]: Large Concept Models: Language Modeling in a Sentence Representation Space

**Questions:**

- do you train a seperate auto-encoder for different values of K ?
- how does CALM perform on in-context recall tasks?

---

### Official Review · Reviewer_Wiky · 2025-11-04

**Soundness:** 3
**Presentation:** 3
**Contribution:** 3
**Rating:** 2
**Confidence:** 3

**Summary:**

The paper proposes to model language "hierarchically" by auto-encoding sentence chunks to a latent sequence which is then modeled with an energy transformer.  This allows for likelihood-free language model training.

**Strengths:**

- The method goes beyond standard next-token prediction and allows for hierarchical modeling of language.
- The proposed methodology for training and evaluation is sound.

**Weaknesses:**

See questions.

**Questions:**

- I'm not sure what the energy interpretation is really doing. The generative head can be thought of as a simple part of the decoder with auxiliary random variables yes? To be clear, an energy function allows for gradient-based sampling which is not done, there is no contrastive divergence or integration during training. What you have seems to be a multi-stage decoder that handles dependencies across the sequence and the generative head is trained via "supervision" through reconstruction. Also, you did sort of choose a likelihood when you chose the squared loss: it's a gaussian log-likelihood. Maybe fewer mentions of likelihood-free training will be useful.


- With brier LM, because you're monte-carlo estimating for each token (unlike log-likelihood which is directly evaluated), you have to produce error bars.

- Where are non-LM evaluations? Like question-answering, retrieval, and long-context understanding (see benchmarks here: https://arxiv.org/abs/2402.18668) ? Brier-LM being correlated with cross-entropy for standard transformers is not sufficient to justify that you trained a good language model, that too on WikiText-103. Further language dataset evaluations are also necessary.

- The authors do not perform sufficient comparisons and related work needs to be refined. You point out that large concept models and MegaByte both face challenges and say that diffusion based generation is slow, but there was no comparison to see how slow. You also say their Auto-encoder is "computationally heavy and fragile" but you also use an auto-encoder, with the main difference being chunking.

---

### Meta-Review · Area_Chair_KX6Z · 2025-12-20

**Summary:**

The paper proposes an alternative language modeling method that predicts continuous representations of tokens (instead of next-token probabilities). The aim of the paper is to improve efficiency via a multi-token modeling approach that is likelihood-free. The reviewers generally find the idea interesting, well written, and conceptually sound, with potential benefits in computational efficiency.

However, all reviews highlight major shortcomings in the evaluation and the discussion of prior work. In particular, the paper lacks standard language modeling benchmarks (e.g., QA, reasoning, long-context recall) and relies too heavily on Brier score correlations, which are seen as insufficient. Comparisons to existing multi-token and hierarchical methods are also missing or underdeveloped. Several reviewers question whether the approach is truly "likelihood-free" or meaningfully energy-based, suggesting the framing may be overstated.

Overall, reviewers view the work as promising but they lean towards rejecting the paper for now. Unfortunately, the authors did not provide any answers, and I therefore have to recommend rejection. I hope the feedback from the reviewers will help the authors improve the manuscript.

**Reviewer Concerns:**

N/A, no rebuttal provided.

**Reviewer Scores:**

N/A

---

### Decision · Program_Chairs · 2026-01-26

Reject